# The Comparison of Latero-Medial versus Dorso-Palmar/Plantar Drilling for Cartilage Removal in the Proximal Interphalangeal Joint

**DOI:** 10.3390/ani11061838

**Published:** 2021-06-21

**Authors:** Alessandro Spadari, Giulia Forni, Sara Del Magno, Claudio Tagliavia, Marco Canova, Annamaria Grandis, Riccardo Rinnovati

**Affiliations:** Department of Veterinary Medical Sciences, University of Bologna, 40064 Bologna, BO, Italy; alessandro.spadari@unibo.it (A.S.); sara.delmagno@unibo.it (S.D.M.); claudio.tagliavia2@unibo.it (C.T.); marco.canova4@unibo.it (M.C.); annamaria.grandis@unibo.it (A.G.); riccardo.rinnovati2@unibo.it (R.R.)

**Keywords:** horse, proximal interphalangeal joint, arthrodesis, cadaveric study

## Abstract

**Simple Summary:**

Arthrodesis of the proximal interphalangeal joint consists of the assisted fusion of the proximal and middle phalanges. The main indications for performing arthrodesis in equine patients are chronic osteoarthritis unresponsive to medical treatment, articular fractures, luxation and subluxation. This procedure can allow a return to athletic career in selected cases, or free the animal from chronic pain in others. Arthrodesis is performed through two basic steps: articular cartilage removal and bone immobilization. Whereas several methods have been studied to achieve the second one, little has been investigated for cartilage removal. The most utilized technique consists of disarticulating the joint to remove the cartilage. Other techniques have been investigated to remove enough cartilage to allow bone fusion and reduce the invasiveness of the procedure. The aims of this work were to assess the capability of a lateral drilling approach to the joint to remove a sufficient amount of cartilage, and compare it to the previously proposed dorsal drilling approach. The lateral drilling approach, especially when performed under digital fluoroscopy, turned out to be more efficient in articular cartilage removal in the proximal interphalangeal joint.

**Abstract:**

The aims of the present study were to compare the percentages of articular cartilage removed using a lateral drilling approach of the proximal interphalangeal joint (PIPJ) and a dorsal drilling approach, and to assess the usefulness of digital fluoroscopy when performing a lateral drilling approach. Sixty cadaveric PIPJs were drilled using a surgical drill bit to remove the articular cartilage. The limbs were divided into three groups containing 10 forelimbs and 10 hindlimbs each. One group received the dorsal drilling approach, the second one received the lateral drilling approach and the last one received the lateral drilling approach under digital fluoroscopy guidance. The percentage of articular cartilage removed from each articular surface was assessed using Adobe Photoshop ^®^ software. The percentages of removed cartilage turned out to be significantly higher with lateral approach, especially under fluoroscopic guidance, both in the forelimbs (*p* = 0.00712) and hindlimbs (*p* = 0.00962). In conclusion, the lateral drilling approach seems to be a minimally invasive technique with which to perform PIPJ arthrodesis, even more efficient than the previously reported dorsal approach.

## 1. Introduction

Chronic osteoarthritis (OA) of the proximal interphalangeal joint (PIPJ) is a common cause of lameness in all types of horses; it can lead to a decrease in athletic performance in sport horses and, ultimately, to a poor quality of life [1,2]; however, no medical treatment has shown long-term satisfactory results [3]. Due to the high load-low motion nature of the PIPJ, facilitated ankylosis or surgical arthrodesis have been proven to eliminate joint motion and the pain associated with it, allowing variable percentages of horses to completely return to their previous athletic performance or pasture soundness [4,5].

Proximal interphalangeal joint arthrodesis has been used in equine medicine for the treatment of debilitating chronic OA and other diseases severely involving the joint, such as articular or comminuted fractures of the proximal and the middle phalanx (P1 and P2, respectively), luxation and subluxation of the PIPJ, osteochondritis dissecans, subchondral bone cysts and end-stage OA resulting from septic arthritis [6,7,8].

Articular cartilage removal and subsequent internal rigid fixation are the two main surgical steps necessary to successfully perform PIPJ arthrodesis [2]. Over past decades, multiple biomechanical and clinical studies have been carried out to evaluate the surgical techniques and implants for performing PIPJ arthrodesis, ranging from using only the transarticular placement of screws having various sizes [7,9,10,11,12,13,14,15] to transarticular screw placement in association with one or two surgical plates of variable designs [1,16,17,18,19,20,21,22,23]. In addition, in selected cases, the technique can also be performed with the horse in a standing position [24]. Nevertheless, very little research has been carried out to investigate the effect of the best cartilage removing technique [25].

Traditionally, PIPJ cartilage removal is performed using an open approach [1,26]. In order to reduce surgical costs and time, and postoperative complications, several less invasive medical and surgical techniques have been investigated. Non-surgical methods include electrical stimulation [27] and the intra-articular injection of monoiodoacetate [28], which have been reported to be scarcely successful due to the remaining articular cartilage within the joint and also because of the irritant effect of these substances on the periarticular soft tissues [28,29], and intra-articular injection with ethyl alcohol, which has shown discordant results [4,30]. Another technique proposed for cartilage debridement consists of using laser diode energy, as previously described for ankylosis of the distal tarsal joint, with encouraging results obtained for both the tarsal and the PIP joints [29,31]. 

Successful ankylosis using stiff joint immobilization without articular cartilage removal has been experimentally and clinically proven in animals other than horses [32,33]. Although a genetic predisposition for naturally occurring ankylosis has also been suggested in horses [34,35], several authors have recommended wide articular cartilage removal, ranging from 30% to complete ablation. [5,36,37,38]. Other authors have suggested that, in PIPJs affected with advanced OA, cartilage removal is not mandatory since a significant portion of articular cartilage has already been destroyed by the pathological process [2,24]. In fact, in another study carried out on healthy animals, it has been reported that bone fusion did not occur in those horses which did not undergo cartilage debridement, even when using different surgical techniques [39]. Therefore, it seems significant to remove as much cartilage as possible, using the least invasive approach.

With the aim of reducing invasiveness and not reducing stability of the joint, a technique for articular cartilage removal that consists of approaching the joint dorsally, sparing the collateral ligaments, has been evaluated. This procedure is less invasive, reduces surgical duration and seems to provide enough cartilage removal to achieve joint fusion [2,25,40].

In the 1990s, a similar minimally invasive technique of cartilage debridement, which consisted of drilling articular surfaces from a lateral approach to the joint under digital fluoroscopy, had already been applied. All the horses treated with this technique returned to their intended use [41].

The authors’ hypothesis was that the technique previously proposed by Bignozzi and colleagues (1993) [41] would remove an appropriate amount of articular cartilage, while being less invasive than other approaches and more rapid in execution. In addition, it was hypothesized that a lateral approach would allow the drill to have more extensive access to the articular surfaces. Therefore, the aim of the present study was to compare dorsal, lateral and lateral under digital fluoroscopy approaches to articular cartilage removal in the PIPJ.

## 2. Materials and Methods

### 2.1. Animals

Sixty distal limbs including metacarpal or metatarsal bone, 30 forelimbs and 30 hindlimbs, were obtained from horses that had been euthanized or had died for reasons unrelated to this study or from orthopedic diseases, the bodies of which were donated for teaching and/or research following written owner consent. The horses were all adults and weighed between 450 and 600 kg. Radiographs of the PIPJ were taken to exclude any affection of the joint. All the procedures were performed by the same operator (A.S.), a skilled Equine Veterinary Surgeon, who had previously practiced the selected techniques in order to standardize the procedures using cadaveric specimens, before collecting the samples included in the study. The subsequent measurements were carried out by three other operators. Both the forelimbs and the hindlimbs were then divided randomly into three equal groups containing 20 limbs each, using a sealed envelope system: dorsal drilling approach, lateral drilling approach and lateral drilling approach under digital fluoroscopy.

The limbs were first clipped from the metacarpo/metatarso-phalangeal joint distally. A DePuy Synthes Colibri II with a 4.5 drill bit was used for both the dorsal and the lateral approaches, as previously described by Bras and colleagues (2011) [25]. 

The duration of each procedure was timed.

### 2.2. Dorsal Drilling Approach

This technique was performed reproducing the one described by Bras and colleagues [25]. During the training sessions, seven drillings were attempted; nevertheless, it was difficult to execute the number of trephinations as indicated [25], due to the small medio-lateral width of the joint. Therefore, only six drillings were performed on each joint. Every limb was positioned on the surgical table with the dorsal surface placed upward; the joint space was detected using percutaneous intraarticular needle insertion. After the needles were correctly positioned in the articular space, at a distance of 1–1.5 cm from each other, a vertical stab incision with an 11 surgical blade of approximately 1–2 cm was made for each needle through the cutis, subcutis, common digital extensor tendon (where present) and articular capsule until the articular space was reached. The needles were then removed, and the drill bit was inserted through the incisions at an angle of 80–84°, as measured each time with a goniometer, to the dorsal surface of the joint (Figure 1a). As the drill bit progressed between the two articular surfaces, the angle was slowly increased to 90°, following the natural curvature of the joint. Each drilling had to go in a straight dorso-palmar/plantar direction and continued as far as possible, limiting damage to the subchondral bone component under the articular surfaces.

### 2.3. Lateral Drilling Approach

This technique was performed reproducing the one described by Bignozzi and colleagues [41], without the aid of digital fluoroscopy. Each limb was positioned on the surgical table with its lateral surface turned upward. The articular space was then identified by palpation and arthrocentesis with the use of two needles; the first was inserted cranially to the lateral collateral ligament and the second was inserted caudally to the ligament to define the preferred site of skin incision and drill bit insertion. A vertical stab incision 1–2 cm long was made using an 11 surgical blade for each needle position to visualize the underlying structures. A sleeve guide was inserted through the incision and the drill bit was driven through it into the joint, so as not to damage the collateral ligament while performing the cranial drilling, and the palmar/plantar digital artery, vein and nerve while performing the caudal drilling. Once the drill bit entered the joint, it was pushed through the articular space, parallel to the articular surfaces, until the medial side of the joint was reached, which was then assessed by digital palpation (Figure 1b). The tip of the drill bit was then moved on the articular plane, in a dorso-palmar/plantar path, keeping the entry point as a fulcrum, until the drill bit was moving freely on its path. Forward and backward movements were also performed with the drill in order to promote the removal of cartilage fragments from the articular space.

### 2.4. Lateral Drilling Approach under Digital Fluoroscopy

This technique was performed by completely reproducing that described by Bignozzi and colleagues [41]. The limbs were approached as previously explained. However, before performing the drilling, it was determined using digital fluoroscopy that the needle, the blade and the drill bit were in the correct position. The directions and depth of the drillings were checked several times in both the latero-medial and the dorso-palmar/plantar projections as the drill bit progressed into the joint (Figure 2a).

### 2.5. Measurements

Once the drilling sessions were concluded and the integrity of the surrounding structures was evaluated (Figure 2c), the soft tissues were dissected and removed in order to expose both the proximal and the distal articular surfaces (Figure 3 and Figure 4), which were then orthogonally photographed using a Fujifilm HS50 next to a ruler. The images were then analyzed using Adobe Photoshop CS 6 Extended^®^ software (Adobe Systems, San Jose, CA, USA) in order to measure in cm^2^ the areas of the total articular surface and of the surface area denuded of cartilage. Using the “freeform pen” tool, the entire articular surface was outlined, obtaining a closed path. The latter was saved as a form, on a new level. Using the same tools, a new layer containing the shape of the area without cartilage was created. 

Subsequently, taking advantage of the ruler included in the picture, the measuring scale of the picture was set up, entering the logical length and logical units in the “Measurement Scale” dialog box (Image > Analysis > Set Measurement Scale > Custom).

Using the “Magic wand” tool, those surfaces where the area had to be measured were then selected. Once the surfaces were selected, the areas in cm^2^ were obtained selecting “Image > Analysis > Record Measurements” tools and opening the “Measurement Log panel”.

### 2.6. Statistical Analysis

The values thus obtained were then processed using Microsoft Office Excel^®^ (Microsoft Excel 2016, Microsoft Corporation, Redmond, WA, USA) to convert the areas into percentages, calculating the amount of articular surface removed.

Descriptive statistics were reported. Given the non-normal distribution of the population, the non-parametric test of Kruskal-Wallis was used to compare the percentages of cartilage removed with the lateral approach drilling under digital fluoroscopy versus the other two methods in both the forelimbs and the hindlimbs.

## 3. Results

The dorsal approach was challenging to perform due to the narrow dorsal articular space. The duration were 12 ± 2 s each, while the duration of the lateral drillings were 45 ± 5 s each. 

No significant differences were found between the cartilage removed from the distal articular surface of the P1 and the proximal surface of the P2. 

The percentages of the cartilage removed from the distal aspect of the P1 and the proximal surface of the P2 in the forelimb were: 27 ± 8% (mean ± standard deviation) with the dorsal drilling approach, 34 ± 12% with the lateral drilling approach and 45 ± 12% with the lateral drilling approach performed under digital fluoroscopy whereas the percentages of the cartilage removed in the hindlimbs were 28 ± 6% with the dorsal drilling approach, 41 ± 9% with the lateral drilling approach and 64 ± 11% with the lateral drilling approach performed under digital fluoroscopy (for details, see Table 1 and Table 2).

The Kruskal-Wallis test showed a *p* value of 0.00712 for the forelimbs and a *p* value of 0.00962 for the hindlimbs.

## 4. Discussion

The aim of the present study was to compare three minimally invasive approaches to articular cartilage removal from the PIPJ by performing a surgical arthrodesis dorsal drilling approach (previously proposed by Bras and colleagues, 2011), lateral drilling approach and lateral drilling approach under digital fluoroscopy (previously proposed by Bignozzi and colleagues, 1993). Our hypothesis was that the lateral approach, especially under digital fluoroscopy, would allow a larger amount of articular cartilage to be removed. A non-parametric test of Kruskal-Wallis was used to compare the percentage of removed articular cartilage by the lateral approach under digital fluoroscopy with the other two methods. The obtained *p* values of 0.00712 for the forelimb and 0.00962 for the hindlimb indicate, respectively, a significant and a highly significant greater amount of cartilage removed by the lateral approach under digital fluoroscopy, compared to lateral and dorsal approaches.

As reported by Bras and colleagues (2011) [25], the removal of 35% of articular cartilage would be sufficient to let the P1 and the P2 fuse; therefore, our lateral approach technique seems to be adequate for performing arthrodesis, except when used in the forelimb without the aid of digital fluoroscopy. Concerning the dorsal drilling approach, in the present study, only 28 ± 6% of the articular cartilage was removed in the hindlimb and 27 ± 8% in the forelimb, indicating that this technique would not allow sufficient removal of the cartilage to execute arthrodesis, as reported by Bras and colleagues (2011) [25] when performing six drillings. Performing seven drillings, at a distance of 1 cm in the dorsal surface of the joint proved to be challenging due to its insufficient latero-medial width. In fact, other authors who reproduced Bras and colleagues technique report the unfeasibility of performing seven drills in all joints [42]. Moreover, changing the declivity of the drill bit from 80–84° to 90° (Figure 5a) during the drillings was not always adequate to precisely follow the rounded articular surfaces of the phalanges, resulting in penetration of the subchondral bone of P2, which could have led to inflammation, necrosis, pain and delayed fusion of the bones in a living animal [39]. However, when comparing the images in this study to those of Bras and colleagues (2011) [25], it could be observed that the seventh drilling tract was barely seen, and was not very evident, in the proximal surface of the P2, thus making the two images very similar. Moreover, the dorsal drillings infrequently reached the palmar/plantar side of the joint in either image. This limitation of the dorsal approach was also noted by Lore and colleagues (2014) [42], who reproduced Bras and colleagues’ technique in their work.

Approaching the joint dorsally allows only one-way drilling, in a dorso-palmar/plantar direction, limiting removal of the cartilaginous tissue to straight strips, whereas when using a lateral approach, although only two points of penetration are required, the drilling can be moved in both a latero-medial pattern and a dorso-palmar/plantar pattern, enlarging the area of removal (Figure 5b).

The anatomic features of the PIPJ, consisting of an articular concave fossa with a central ridge in the middle phalanx, and in two convex condyles in the proximal phalanx, do not impede the latero-medial progress of the drill bit since the drill bit can remain aligned with the articular edge through the entire pathway, unlike what happens when using a dorsal drilling approach. In fact, due to the natural curvature of the proximal articular surface of P2 and the resulting requirement to change the drilling angle during the procedure, a dorsal drilling approach could increase the risk of damaging the subchondral bone, which would cause more pain and delay in healing [39].

So far, to perform an arthrodesis of the PIPJ, arthrotomy with total removal of the articular cartilage has been the most recommended technique [3,10]. Although this procedure is very effective, it is also particularly invasive; wide exposure of the articular surfaces and surrounding soft tissues as well as the transection of the common digital extensor tendon, and of the medial and lateral collateral ligaments predispose to sepsis, and prolong postoperative progress and healing processes. More recently, less invasive techniques to approach the joint have been designed [2,21,25,28,29,30,42]. These procedures allow a reduction in operating time (which is very important in equine surgery since prolonged recumbency may lead to ischemic myopathy), a decrease in postoperative pain and hospitalization time, an increase in joint stability, better cosmetic result and, finally, a reduction in operative costs [25].

Most of these less invasive techniques consist of removing as much articular cartilage as possible by drilling the joint surfaces without disrupting the joint, similar to the technique previously designed for the distal tarsal joint [37]. In fact, the disarticulation of a joint, especially transection of the collateral ligaments, leads to lower stability of the joint, as reported in human medicine regarding ankle joints [43]. Moreover, when performing arthrodesis of the metacarpo/metatarso-phalangeal joint, a condilectomy of the distal aspect of the third metacarpus/metatarsus is recommended instead of transection of the collateral ligaments in order to provide higher stability to the arthrodesed joint [44]. The main limitation of less invasive techniques is the difficulty in determining whether enough cartilaginous tissue has been removed to allow bony fusion. For this reason, cadaveric studies, such as the present one, are key for subsequent clinical applications.

A lateral minimally invasive approach to achieve arthrodesis of the PIPJ has been described also by Lescun (2008) [45]; however, this technique was not considered significant by subsequent authors due to the risk of damaging the collateral ligaments [25]. In addition to the collateral ligaments, the palmar/plantar digital vein, artery and nerve could be damaged by drilling if the drill was inserted too caudally, and the dorsal branch of the palmar/plantar digital nerve could also be damaged if the drill bit was inserted too cranially. However, this eventuality can be avoided by making a longer incision through the cutis and using a sleeve guide to shift the structures away from the drill insertion site. Moreover, the orthopedic drill bit used by Lescun was a 5.5 mm bit while a thinner drill bit was used in this study, which was less damaging to the subchondral bone and the periarticular structures. In fact, the use of 5.5 drill bit was not subsequently recommended because it resulted in excessive bone removal [25,40]. Nevertheless, Lescun’s results were quite satisfactory, even considering the small number of cases, with five of the six horses found to be healthy six months postoperatively [45]. It can be hypothesized that, as Kuemmerle and Berchtold (2013) [40] have reported, using a Spratt curette would be less damaging, for both soft and bony tissues, than using a drill bit. However, a drill bit may be more maneuverable inside the joint space and may allow treating a larger area. 

Some authors have proposed a combination of dorsal and lateral approaches and have compared it to Bras and colleagues’ technique [42]. This technique consisted in performing seven dorsal drilligs and one lateral or medial drilling in the palmar/plantar part of the joint space, in order to compensate for the lack of cartilage removal in this particular side of the joint, achieved with the dorsal approach alone. Lore and colleagues (2014) [42] managed to remove 45 ± 5% of articular cartilage with their technique, which was significantly higher than that obtained by reproducing Bras and colleagues approach (34 ± 4%). This supports the hypothesis that a lateral approach to the PIPJ is necessary to remove a larger amount of cartilage.

In the present study, all the drillings were performed by the same operator in order to reduce the variables which could influence the measurements. The duration of each drilling was timed in order to provide an accurate description of the techniques. Each drilling continued until the drill bit could move in the selected course through the joint space without resistance. The differences encountered between the duration of the dorsal drillings (12 ± 2 s) and the lateral drillings (45 ± 5 s) was ascribed to the different paths that the drill bit had to follow; in fact, when approaching the joint dorsally, the anatomy of the phalanges allows only a straight path to their caudal side. Otherwise, too much subchondral bone would be damaged; thus each drilling lasted only a few seconds whereas, using a lateral approach, having the chance to move the drill bit in both straight and dorso-palmar/plantar movements, increased the duration of the drilling.

Another interesting result found in the present study was the difference in the percentage of cartilage removed between the forelimb and the hindlimb, which appeared to be significant in all the techniques performed in the present study. In addition, what has been reported in the literature is in complete agreement with the fact that the success rate for PIPJ arthrodesis surgery in horses is higher in the hindlimbs than in the forelimbs [1,3,5,10,36,46]. It could be hypothesized that this difference in results between the forelimbs and the hindlimbs could be ascribed to the different convexity and concavity of the articular surfaces of the anterior and the posterior P1 and P2, respectively, which make straight drilling not perfectly matched to make contact with the articular planes, mainly in the forelimb bone extremities. An anatomical difference between forelimb and hindlimb had previously been hypothesized by other authors [42]; nevertheless, to the authors’ knowledge, morphometric studies comparing fore- and hindlimb PIPJ are lacking. However, in the study carried out by Kummerle and Berchtold (2013) [40] on cadaveric specimens in which articular cartilage was removed using a Spratt curette, no difference in the percentage of tissue removed between the forelimb and the hindlimb was found. Instead, a higher percentage of cartilage removed from the proximal phalanx than from the distal phalanx, in contrast with the present results and with the literature was reported [25,40].

In order to obtain successful arthrodesis, it is necessary that the two trabecular, or subchondral, bone surfaces are well aligned, compressed and stabilized [12]. Since the presence of cartilage, during the ankyloses process, interferes with vascular invasion and new bone formation between the proximal and the middle phalanx, as much cartilaginous tissue as possible must be removed in order to allow maximum contact between the bones [47]. Spontaneous ankyloses would eventually occur when a joint is immobilized, even if cartilage is not removed, because fixation leads to malnutrition of the cartilaginous tissue and its subsequent degeneration. However, this process can last for many years, causing pain to the horse. Some authors have stated that in severe cases of OA affecting the PIPJ, removal of articular cartilage is not a mandatory step to achieve successful arthrodesis, suggesting that an advanced degree of arthrosis implies a great loss in cartilaginous tissue and, therefore, stabilization of the joint alone would be sufficient in reducing pain and allowing joint fusion [24]. However, the majority of authors, including those of this paper, disagree with that concept. It has been suggested that the incomplete removal of articular cartilage could lead to asymmetry of the articular planes and also possibly cause loosening of the fixation implants; in fact, the remaining cartilage would deteriorate over time, leaving an empty space inside the joint, which could eventually lead to instability and implant failure [14,17,25].

In cases of severe arthrosis, the presence of fibrosis and osteophytosis surrounding the joint can compromise the anatomy of the bony structures, making it challenging to identify the articular space [46]. This could be particularly noticeable when approaching the joint dorsally [40] whereas a lateral approach would be easier to perform since it requires fewer points of penetration of the joint, and there are landmarks which are more easily identified. This could be even more evident when performed under digital fluoroscopy in which the drill bit can be followed in its entire latero-medial pathway while it would not be useful with a dorsal approach due to the overlapping of the structures. The advantage of using fluoroscopy is to precisely and rapidly detect (from the screen) the exact position of the joint surfaces in which the drill has to be properly inserted to measure the depth of the progression until the opposite (medial) aspect of the joint, and to keep the movement in proper alignment, not advancing into the subchondral or trabecular bone during the progression and the dorso-palmar/plantar movements. Moreover, it helps avoiding damage of extra-articular soft tissues and may reduce operative time. On the other hand, the use of digital fluoroscopy exposes the surgeon to radiation [41].

Since the validity of the lateral approach for the drilling of articular cartilage was verified with the present study, it can be stated that the technique previously performed by Bignozzi and colleagues (1993) [41] was confirmed as a minimally invasive technique to remove articular cartilage, demonstrating its effectiveness. In fact, although only a few cases were presented in that study, they all returned to their intended use. It would be interesting to apply it to a larger number of cases of joints affected with OA in order to validate its effectiveness in a clinical setting. In fact, in very advanced cases of OA is for healing problems impossible to use the open approach and this method in combination with fixation implants may be only reasonable solution to pain originating from pastern joint.

Despite the encouraging results obtained, the study includes limitations: first, the fact that the operator who performed the drillings was not blinded. The operators who executed the Photoshop analysis were also not blinded due to the impossibility of not recognizing the lateral from the dorsal approach. Another limitation of this study was that the percentages of cartilage removed were calculated on two-dimensional photographs. Since the articular surfaces of the proximal interphalangeal joint are convex (distal aspect of the first phalanx) and concave (proximal aspect of the second phalanx), respectively, the measurements could have been slightly underestimated from the real three-dimensional values. Moreover, the depth of the cartilage removed was not assessed histologically.

## 5. Conclusions

This study suggests that removing articular cartilage from the PIPJ by inserting an orthopedic drill from the lateral side of the joint, especially under fluoroscopic guidance, can be considered more efficient than a dorsal approach as a minimally invasive technique for articular cartilage removal.

## Figures and Tables

**Figure 1 animals-11-01838-f001:**
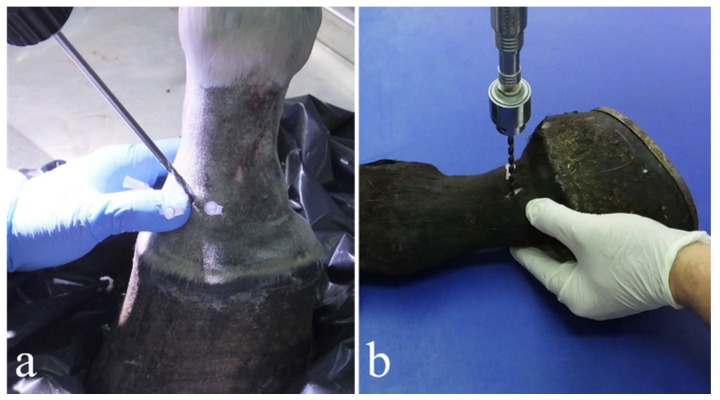
Dorsal (**a**) and lateral (**b**) drilling approach. The articular space was identified by the insertion of a series of hypodermic needles at a distance of 1 cm from each other.

**Figure 2 animals-11-01838-f002:**
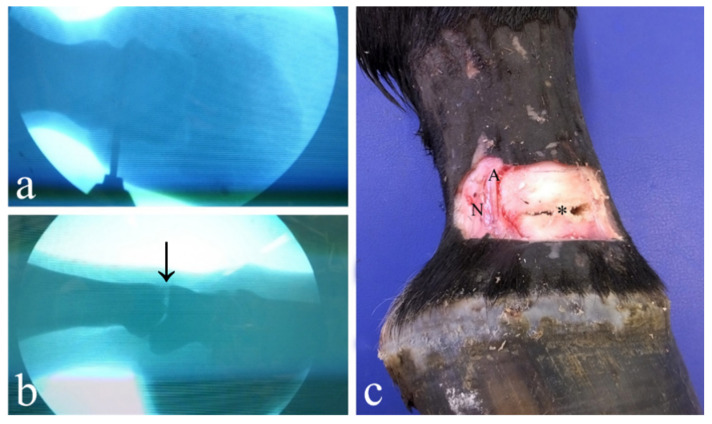
Lateral drilling approach under digital fluoroscopy. The correct drill bit positioning was checked several times by the use of the digital fluoroscopy ((**a**) dorsal view). Note the radiolucent area (arrow) following the articular cartilage removal of the articular surface of PIPJ ((**b**) lateral view) and the drill holes placed dorsally and palmar/plantar to the collateral ligament (asterisk) ((**c**) lateral view). A, arteria digitalis palmaris propria lateralis; N, nervus digitalis palmaris proprius lateralis.

**Figure 3 animals-11-01838-f003:**
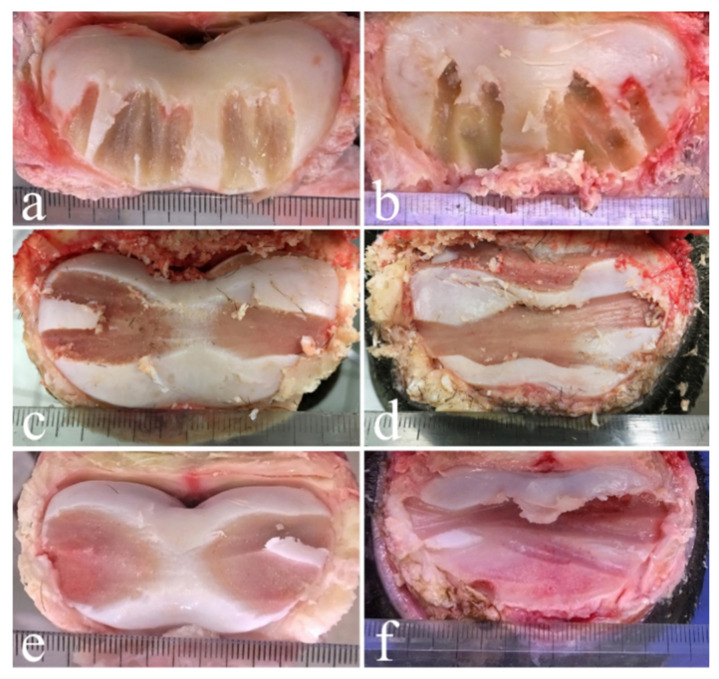
Distal surfaces of P1 (**a**,**c**,**e**) and proximal surfaces of P2 (**b**,**d**,**f**) of the forelimb after dorsal drilling approach (**a**,**b**) and lateral drilling approach without (**c**,**d**) and with the use of digital fluoroscopy (**e**,**f**). The best cases, belonging to three different animals and obtained with the three different drilling techniques, are shown.

**Figure 4 animals-11-01838-f004:**
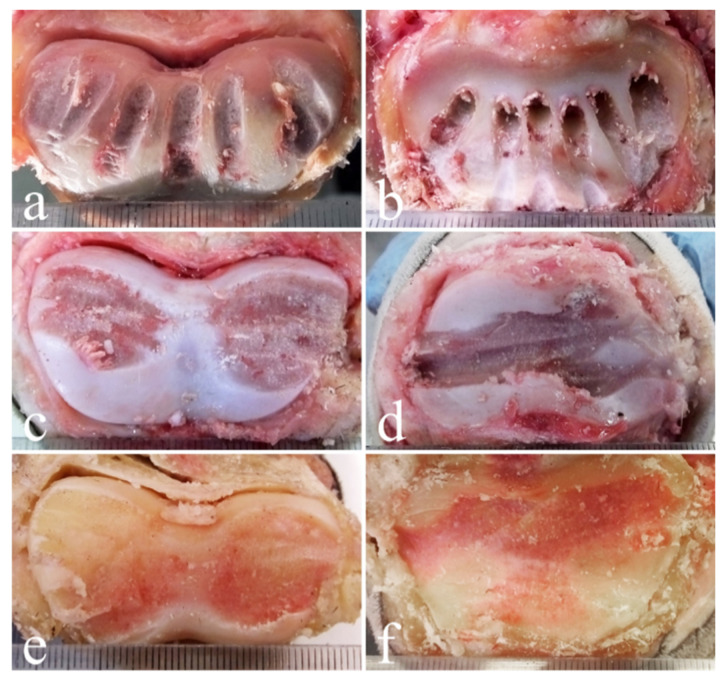
Distal surfaces of P1 (**a**,**c**,**e**) and proximal surfaces of P2 (**b**,**d**,**f**) of the hindlimb after dorsal drilling approach (**a**,**b**) and lateral drilling approach without (**c**,**d**) and with the use of digital fluoroscopy (**e**,**f**). The best cases, belonging to three different animals and obtained with the three different drilling techniques, are shown.

**Figure 5 animals-11-01838-f005:**
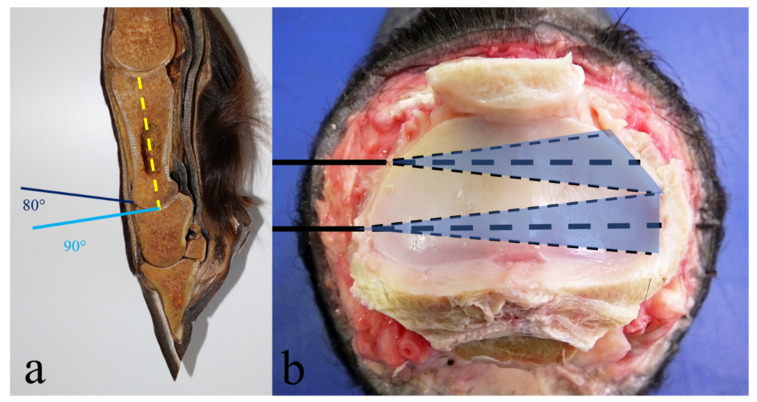
Dorsal drilling approach pattern, showing the entry angle into the joint (80°) and the requirement to change the drilling angle, in order to follow the articular surfaces curvature (**a**). Lateral drilling approach pattern, showing the areas interested by cartilage removal according to the movements of the drill bit (**b**).

**Table 1 animals-11-01838-t001:** Mean ± SD total area (cm^2^), eroded area (cm^2^) and percentage of removed cartilage in proximal and distal articular surfaces of the proximal interphalangeal joint (PIPJ) in the forelimb. Comparison of the three techniques.

Drilling Approach	Forelimb							
	Proximal Articular Surface (Distal P1)			Distal Articular Surface (Proximal P2)			% of Total PIPJ Erosion	*p* Value
	Total Area	Eroded Area	% of Removed Cartilage	Total Area	Eroded Area	% of Removed Cartilage		
Dorsal	12 ± 3	4 ± 1	30 ± 10	14 ± 2	4 ± 1	25 ± 8	27 ± 8	0.00712
Lateral without digital fluoroscopy	12 ± 2	4 ± 2	33 ± 11	13 ± 2	5 ± 2	34 ± 13	34 ± 12	
Lateral under digital fluoroscopy	14 ± 4	6 ± 2	43 ± 16	14 ± 3	6 ± 2	48 ± 20	45 ± 12	

**Table 2 animals-11-01838-t002:** Mean ± SD total area (cm^2^), eroded area (cm^2^) and percentage of removed cartilage in proximal and distal articular surfaces of the proximal interphalangeal joint (PIPJ) in the hindlimb. Comparison of the three techniques.

Drilling Approach	Hindlimb							
	Proximal Articular Surface (Distal P1)			Distal Articular Surface (Proximal P2)			% of Total PIPJ Erosion	*p* Value
	Total Area	Eroded Area	% of Removed Cartilage	Total Area	Eroded Area	% of Removed Cartilage		
Dorsal	11 ± 3	3 ± 7	28 ± 8	12 ± 2	3 ± 1	29 ± 5	28 ± 6	0.00962
Lateral without digital fluoroscopy	13 ± 3	5 ± 1	38 ± 10	13 ± 2	6 ± 2	45 ± 9	41 ± 9	
Lateral under digital fluoroscopy	13 ± 3	8 ± 2	61 ± 9	12 ± 3	8 ± 3	61 ± 24	64 ± 11	

## Data Availability

The data present in this study are available in the article.

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
