# Peer review of "The Comparison of Latero-Medial versus Dorso-Palmar/Plantar Drilling for Cartilage Removal in the Proximal Interphalangeal Joint"

_animals, 2021, doi:10.3390/ani11061838_

Round 1

Reviewer 1 Report

Review of Animals 1206179

Overall a very well designed study that addresses an important issue.  The introduction and methods are clear and concise; however, the results need editing and the discussion needs complete revision as it is too long and scattered.

Materials and methods

Line 104 – ‘Radiographies’ needs to be changed to ‘radiographs’

Line 118 – First sentence is a repeat of the above sentence (Line 115)

Line 121 – ‘Size’ needs to be changed to ‘width’

Line 128 – Replace ‘these little incisions with’ phrase with ‘the incisions at’

Line 142 – Replace ‘to show and recognize’ with ‘to visualize’

Line 147 – Replace ‘phalangeal’ with ‘articular’

Figure 1 – In the dorsal approach “inserting a few needles at a distance of 1 cm” was used, but not the lateral approach.  Replace ‘inserting a few needles’ with ‘the insertion of a series of hypodermic needles’

Figure 2 – Replace ‘right direction of the drills’ with ‘correct drill bit positioning’.  Missing arrow and asterisk in 2b.  Replace ‘erosion’ with ‘articular cartilage removal’. Replace ‘caudally’ with palmar or plantar if visible.  I only see defects dorsal to the collateral ligament in 2c.  Missing labels A and N in 2c.

Line 175 – It is unclear what specifically is referred to in Figure 2b as it does not show any image of soft tissues being “checked”.  Replace ‘checked’ with ‘evaluated’

Line 176 – Figure 2c shows soft tissue dissection, but the reader to referred to Figures 3 and 4 instead, which illustrate disarticulation and visualization of the articular surfaces and not soft tissue dissection.

Lines 179-188 – I do not see the value of the added detail of the steps within the software menu to collect the area of the defects.

Figure 3 – The labels a-f need to be enlarged and changed to white font to improve visibility. Replace ‘types of drilling’ with drilling technique’ or something similar.

Figure 4 – Same comments as for Figure 3.

Line 204 – Replace ‘carried out’ with reported.

Lines 204-206 – Combine these two sentences as “The test’ is unclear (Lines 205-206).

Line 208 – Replace ‘dorsal drillings’ with ‘dorsal approach’.  Be consistent in terminology throughout.  Replace ‘turned out to be challenging’ with ‘were challenging’.

Lines 209-210 – Replace ‘they continued’ with ‘the duration’

Lines 210-211 – The area of debrided cartilage is clearly different for the proximal versus distal articular surfaces within an articulation, yet the proximal and distal surfaces seem to be pooled here.  Put this data in a table and do not report clinically irrelevant significant digits (27.07%) in the results when 27% is correct.

Line 218 – Unclear what the Kruskal-Wallis p-values represent.  Need to provide a much clearer explanation for the reader of what comparisons were done.

Line 220 – If indeed there were no significant differences between the proximal and distal defects, then this needs to be reported first before reporting the pooled values (see Lines 210-211 above).

Figures 3 and 4 – So it seems that “best cases” is not within the same horse.  This needs to be better described in the figure legends that the proximal and distal surfaces shown are all from different horses.

Table 1 – Data is unreadable as presented.  Remove significant digits and divide into 2 separate tables for forelimbs and hind limbs.  Need to add superscripts of significant group differences from the Kruskal-Wallis tests.

Discussion – The first paragraph needs to present the primary results of the paper and the hypothesis testing.  This is a restatement of the introduction.  Delete this paragraph.  The discussion is too long and rambling and poorly organized.

Line 243 – Is it ‘trabecular’ bone or can exposure of the subchondral bone have the same effect?

Line 244 – Cartilage does not ‘inhibit’ from a physiologic perspective.  Be clear that you are referring to the surgical procedure and the debridement of cartilage.

Line 250 – Remove ‘much’ pain

Lines 250-253 – Reword sentence as this is confusing and scattered.  Rework this entire paragraph as it is not consistent in its flow of logic and includes scattered concepts.

Line 271 – Be consistent in terminology – ‘sparing’ versus ‘saving’

Lines 272 – 41% and 80%.

All paragraphs in the discussion need reorganization and condensing.  Scattered concepts and repeated topics are very confusing.  Each paragraphs needs to address a one specific topic – amount of cartilage removal; surgical approaches; invasiveness of procedures; use of fluoroscopy, etc.

Line 305 – The aim of the study was not to assess the relevancy of the lateral approach.  It was to compare the three different techniques.

Lines 315-320 – Restating results – delete paragraph.

Line 322 – Now talking about cartilage removal again…  Move and condense material.

Figure 5 – Move to results.

Line 264 – Did any of your cases have severe osteophytosis?

Line 377 – Be consistent in terms – “DJD” versus “OA”

Line 379 – And now discussing techniques again… Move and condense material.

Line 410 - And cartilage removal again… Move and condense material.

Add subheading for ‘Limitations’ and condense material.

Line 454 – Conclusion is not accurate.  Need to conclude what your study actually tested.  You did not test disarticulation of the joint.

Reviewer 2 Report

This is a partially innovative contribution that compares previous studies and hypothesis.The originality of the work and the repercussions on clinical applications should be better clarified.

Reviewer 3 Report

I would like to stress one aspect of this method of cartilage destruction - in very advances cases of high ringbone is for healing problems impossible to use the open approach and this method in combination with simple percutaneous application of lag srews is only reasonable solution of the pain originating from pastern joint. 
